# Depletion of kinesin motor KIF20A to target cell fate control suppresses medulloblastoma tumour growth

Runxiang Qiu[1], Jun Wu[2], Brian Gudenas[3], Paul A. Northcott [3], Robert J. Wechsler-Reya [4] & Qiang Lu [1✉]

During mammalian brain development, neural progenitor cells proliferate extensively but can ensure the production of correct numbers of various types of mature cells by balancing symmetric proliferative versus asymmetric differentiative cell divisions. This process of cell fate determination may be harnessed for developing cancer therapy. Here, we test this idea by targeting KIF20A, a mitotic kinesin crucial for the control of cell division modes, in a genetic model of medulloblastoma (MB) and human MB cells. Inducible *Kif20a* knockout in both normal and MB-initiating granule neuron progenitors (GNPs) causes early cell cycle exit and precocious neuronal differentiation without causing cytokinesis failure and suppresses the development of Sonic Hedgehog (SHH)-activated MB. Inducible KIF20A knockdown in human MB cells inhibits proliferation both in cultures and in growing tumors. Our results indicate that targeting the fate specification process of nascent daughter cells presents a novel avenue for developing anti-proliferation treatment for malignant brain tumors.

[1] Department of Developmental and Stem Cell Biology, Beckman Research Institute of the City of Hope, Duarte, CA, USA. [2] Division of Comparative Medicine, Beckman Research Institute of the City of Hope, Duarte, CA, USA. [3] Department of Developmental Neurobiology, St. Jude Children's Research Hospital, Memphis, TN, USA. [4] Tumor Initiation and Maintenance Program, National Cancer Institute–Designated Cancer Center, Sanford Burnham Prebys Medical Discovery Institute, La Jolla, CA, USA. ✉email: qlu@coh.org

During mammalian brain development, extensive expansion occurs in neural progenitor cells (NPCs); however, in contrast to the proliferating cells in a growing tumor, NPCs can produce correct numbers of diverse cell types without running a risk of excessive or dysregulated proliferation. NPCs achieve this feat through coordinated actions of growth-stimulating and -suppressing factors leading to symmetric or asymmetric cell divisions, which endow a proliferative or differentiative fate to daughter cells, so that an overall balance between proliferation and differentiation is ensured throughout brain morphogenesis. Disruption of the balance between symmetric and asymmetric cell divisions is expected to cause an imbalance between proliferation and differentiation, and may lead to tumorigenesis. Genetic studies in the *Drosophila* nervous system presented early evidence linking defects in cell division mode to tumorigenesis. In the *Drosophila* neuroblasts, the fly NPCs, mutations of multiple regulators of asymmetric cell division have been linked to hyperproliferation of neuroblasts in the embryos[1–6] and tumor-like growth in recipient flies when mutant cells were transplanted[1]. These observations indicated that misregulation of the fate of an NPC's daughter cells could lead to cancer[7,8]. These observations also suggested that the fate specification process of daughter cells might serve as a point of intervention for developing cancer therapy. However, key regulators that control symmetric proliferative vs. asymmetric differentiative divisions in the mammalian NPCs have remained largely elusive and, hence, the idea of targeting regulators of cell division mode for therapeutic intervention has not been directly tested in the mammalian systems.

We have recently identified KIF20A, a mitotic kinesin previously implicated in cytokinesis regulation in cancer cells, with a crucial role in balancing symmetric vs. asymmetric NPC divisions during cerebral cortical development[9]. Our data showed that inactivation of KIF20A through either short hairpin RNA (shRNA) knockdown or genetic knockout causes the affected cortical NPCs to switch from proliferative to differentiative mode of divisions. KIF20A exerts its function in the process of daughter cell-fate specification during NPC divisions in coordination with RGS3[9] and SEPT7[10]. As tumor-initiating (stem/progenitor-like) cells share many regulatory mechanisms of cell proliferation with normal neural stem/progenitor cells[11–15], it is conceivable that KIF20A is also crucial for the control of proliferation vs. differentiation of tumor-initiating cells.

In human cancers, KIF20A shows low mutation rate based on available TCGA database, which is consistent with the essential role of KIF20A in cell divisions of NPCs or other stem/progenitor cells (germline knockout of the *Kif20a* leads to embryonic/perinatal lethality). On the other hand, in silico studies of cancer expression data or expression analyses of clinical samples showed that KIF20A expression is positively correlated with poor prognosis of patients in different types of cancers[16–19]. Based on these observations, we anticipate that disruption of KIF20A function in brain tumor-initiating cells would similarly promote the affected cells to leave the cell cycle and become post mitotic, and as a result, this should lead to inhibition of brain tumor initiation and/or progression.

To test this idea, we employed a genetic model of medulloblastoma (MB) induced by sustained activation of the Sonic Hedgehog (SHH) pathway. In this model system, inducible knockout of the *Patched (Ptc)* gene in cerebellar granule neuron progenitors (GNPs) (via expression of CreER under the control of the *Atoh1* promoter) results in MB formation within a few months[20]. As KIF20A is a mitotic protein that should be expressed by NPCs present in other central nervous system areas beyond the cerebral cortex, we expected that KIF20A might play a similar role in regulating cell division mode, both in normal

cerebellar GNPs during development and in tumor-initiating GNPs during tumor formation. To target KIF20A in the cerebellum, we established compound mice carrying *Atoh1-CreER*, *Kif20a^{fl/fl}* and *Atoh1-CreER, Kif20a^{fl/fl}, Ptc^{fl/fl}* alleles. Deletion of the *Kif20a* gene alone or deletion of both the *Ptc* and *Kif20a* genes in cerebellar GNPs was investigated. We will present data obtained from these experiments showing that loss-of-function (LOF) of KIF20A resulted in early cell cycle exit and precocious neuronal differentiation in both normal and tumor-initiating GNPs, as well as data from targeting KIF20A function in human MB cells both in vitro and in vivo.

## Results

**Conditional knockout of *Kif20a* in cerebellar GNPs causes early cell cycle exit and precocious neuronal differentiation.** In the mouse cerebellum, immunostaining showed that strong KIF20A expression is enriched within the external granular layer (EGL), where GNPs are located during early postnatal stages (Fig. 1a). To study the potential function of KIF20A in cerebellar development, we performed inducible deletion of the *Kif20a* gene in GNPs. Conditional *Kif20a*-knockout mice (littermates of *Atoh1-CreER; Kif20a^{fl/fl}* and control *Kif20a^{fl/fl}* mice) were treated with tamoxifen at postnatal day 4 (P4). Brain samples were then collected from P5 for analyses. Assessment of KIF20A expression in the P5 and P6 samples indicated that deletion of *Kif20a* was evident 1 day after tamoxifen induction (Supplementary Fig. 1a). Examination of Ki67 expression in the brain samples revealed a loss of Ki67+ cells in the mutant cerebellums following the induced deletion of *Kif20a* (Fig. 1b and Supplementary Fig. 1b). To address whether the loss of proliferating cells resulted from increased death of mutant GNPs (due to possible defect in cytokinesis after *Kif20a* knockout), we performed immunostaining for activated caspase 3. Our results from P6 and P7 brain samples showed that the apoptosis levels in the cerebellums were overall low and comparable between the *Kif20a*-knockout and wild-type littermate brains (Fig. 1c and Supplementary Fig. 1c). We previously found that knockout of *Kif20a* did not cause a noticeable failure of cytokinesis in neural progenitor divisions of the developing cerebral cortex[9]. To assess the status of cytokinesis process in the mutant cerebellums, we performed flow cytometry-based cell cycle analysis. Our results showed that the 4N DNA contents between the P6 mutant (*Atoh1-CreER; Kif20a^{fl/fl}*) and control (*Kif20a^{fl/fl}*) cerebellar cells were comparable (Fig. 1d), indicating that knockout of *Kif20a* did not cause an obvious defect in cytokinesis of GNP divisions, or a possible cytokinesis defect was largely compensated by other factors.

We next examined the cell cycle exit and re-entry in GNPs of the *Kif20a*-knockout cerebellum. In this experiment, conditional *Kif20a* mice were treated with tamoxifen at P4 followed by an injection of EdU at P5, and brain samples were then collected at P6 or P7 for analyses (Fig. 2). In the P6 brains, co-staining of EdU and Ki67 showed that within individual cerebellar sulci, there were relatively more EdU+Ki67− cells (cells having left the cell cycle after being labeled by EdU) in the mutant EGL than in the wild-type littermate EGL (Fig. 2a). Co-staining with neuronal marker NeuN further showed that more EdU+Ki67− cells in the *Kif20a*-knockout brains were positive for NeuN (Fig. 2b). These EdU+Ki67−NeuN+ cells were located outside the proliferating zone (Ki67+ cell zone) (Fig. 2b), reflecting developmental progression of differentiated granule neurons emanating from the EGL. This result also suggested that knockout of *Kif20a* did not compromise the completion of cell division (cytokinesis) of the mutant GNPs. In the P7 brains, within individual cerebellar sulci, significantly fewer EdU+Ki67+ cells (cells remaining in the cell cycle) could be seen in the mutant EGL than in the wild-type

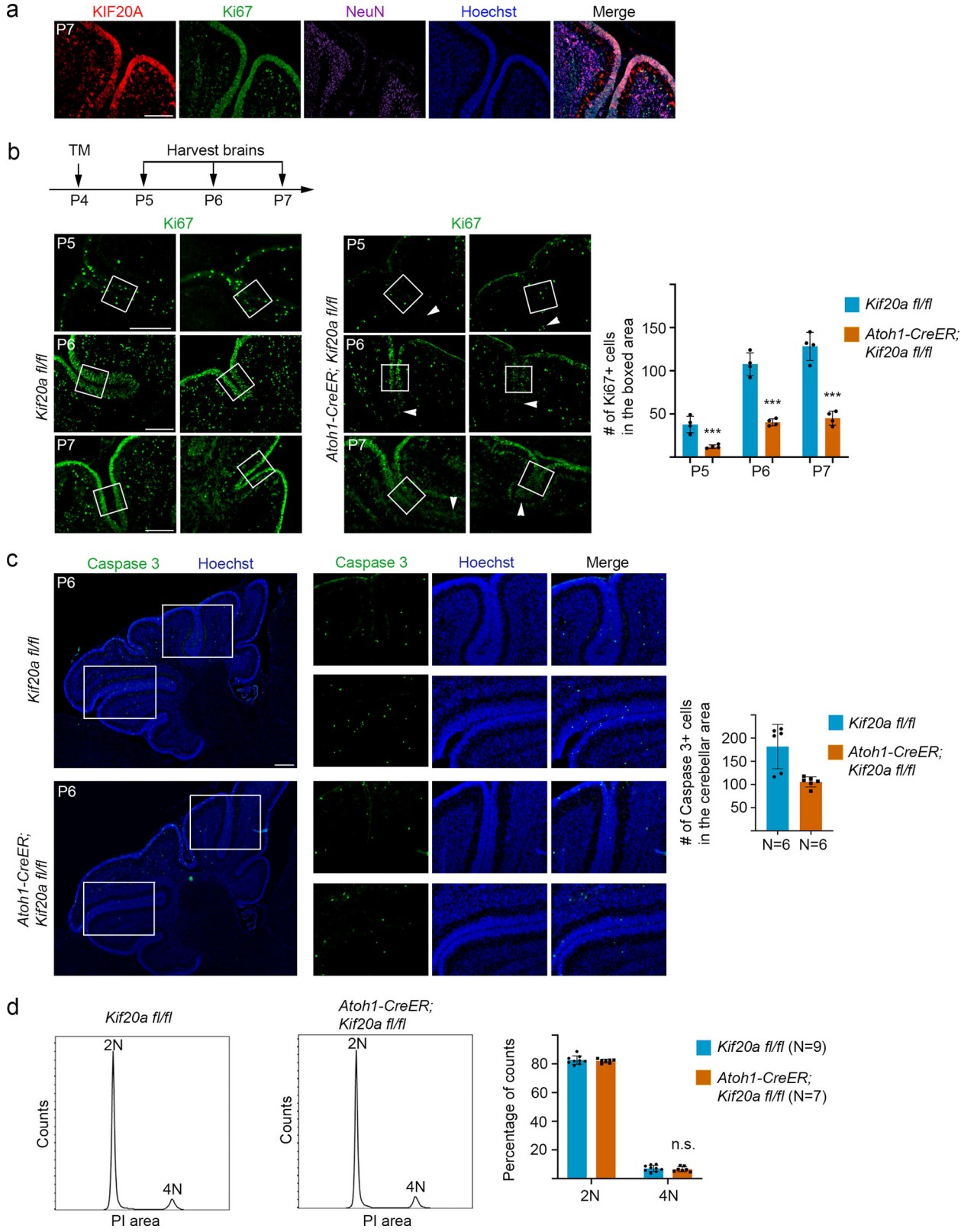

EGL (Fig. 2c). These data collectively suggested that KIF20A is essential for maintaining the proliferative state of GNPs and that LOF of KIF20A in GNPs causes cell cycle exit and leads to an early depletion of the progenitor population.

**Blocking KIF20A function inhibits SHH-induced MB**. To address whether KIF20A might be similarly crucial for

maintaining the proliferation of brain tumor-initiating cells, we crossed the *Kif20a*<sup>fl/fl</sup> floxed mice with *Atoh1-CreER; Ptc*<sup>fl/fl</sup> mice to generate a strain of compound mice carrying the *Atoh1-CreER; Ptc*<sup>fl/fl</sup>; *Kif20a*<sup>fl/fl</sup> alleles. When tamoxifen was given to the pups at P4, *Ptc*-knockout mice with intact *Kif20a* (*Atoh1-CreER; Ptc*<sup>fl/fl</sup>) developed MB with 100% penetrance within about 5 months (Fig. 3a), as previously reported[20]. *Ptc/Kif20a* double-knockout

**Fig. 1 LOF of KIF20A in GNPs results in a loss of proliferating cells. a** KIF20A expression was mainly enriched in the proliferating cell zone in the external granular layer (EGL) of the early postnatal cerebellum. Scale bar represents 50 µm. **b** Tamoxifen (TM)-induced knockout of *Kif20a* in GNPs caused a reduction in the number of proliferating cells in the mutant cerebellums. Ki67$^+$ cells within boxed region at the center of individual cerebellar sulcus were used for quantification. White arrowheads indicated the tips of sulci in the mutant cerebellums, which often had diminished Ki67 signal. Scale bar represents 50 µm. Data are mean ± SD. $P = 0.0019$ (P5), 6.65E − 05 (P6), 9.52E − 05 (P7) (Student's *t*-test). **c** Knockout of *Kif20a* did not cause a noticeable increase of apoptosis level marked by cleaved caspase 3 in the P6 mutant cerebellums. Scale bar represents 50 µm. Data are mean ± SD. **d** Flow cytometry-based cell cycle analysis of P6 littermate of knockout and control cerebellums. The 4N DNA contents (representing G2/M phase cells or cells with two nuclei resulted from cytokinesis defect) were little changed between the *Kif20a*-knockout and control brains. n.s., not significant.

mice (*Atoh1-CreER; Ptc^{fl/fl}; Kif20a^{fl/fl}*), on the other hand, showed a longer survival time before symptoms occurred and some of the mice appeared normal even 1 year after tamoxifen induction (Fig. 3a). Among the mice that developed symptoms, the brains of the *Ptc/Kif20a* double-knockout mice had smaller tumors than those seen in *Ptc*-knockout mice with intact *Kif20a* (Fig. 3b–h). The brain samples from the *Ptc/Kif20a* double-knockout mice generally also retained more normal morphology than those *Ptc* single-knockout mice (Fig. 3b and Fig. 3f–h vs. 3d, e). These results suggest that KIF20A is crucial for the development of SHH-induced MB.

**LOF of KIF20A in MB-initiating GNPs causes early cell cycle exit.** To study the possible mechanism of inhibition of SHH-induced MB by LOF of KIF20A, we looked at cell cycle exit and re-entry in tumor-initiating GNPs of the *Ptc* single- and *Ptc/Kif20a* double-knockout mice. The pups of *Atoh1-CreER; Ptc^{fl/fl}* and *Atoh1-CreER; Ptc^{fl/fl}; Kif20a^{fl/fl}* mice were treated with tamoxifen at P4 followed by EdU labeling at P5. Brain samples were then collected at P6 and P7 for analysis. In the P6 brains, data from Ki67 staining showed that there were overall fewer Ki67$^+$ cells in the cerebellar sulci of the *Ptc/Kif20a* double-knockout brains than those of the *Ptc* single-knockout brains (Fig. 4a). Co-staining of EdU and Ki67 showed that there were more EdU$^+$Ki67$^-$ cells in the *Ptc/Kif20a* double-knockout EGL than in the *Ptc* single-knockout EGL (Fig. 4a), suggesting that more GNPs in the *Ptc/Kif20a* double-knockout brains had left the cell cycle earlier. Co-staining with neuronal marker NeuN further showed that more EdU$^+$Ki67$^-$ cells in the *Ptc/Kif20a* double-knockout brains were positive for NeuN and these EdU$^+$Ki67$^-$NeuN$^+$ cells formed a line surrounding the proliferating zone, reflecting newly born granule neurons exiting the EGL (Fig. 4b). In the P7 brains, fewer proliferating cells (EdU$^+$Ki67$^+$ cells) remained in the EGL of the *Ptc/Kif20a* double-knockout mice than in the *Ptc* single-knockout EGL (Fig. 4c). These data collectively indicate that KIF20A is also essential for maintaining the proliferative state of tumor-initiating GNPs.

**LOF of KIF20A in mouse tumor cells leads to inhibition of proliferation by inducing differentiation.** To further examine the mechanisms of KIF20A function in MB, we sought to isolate proliferating cells from MBs derived from both the *Atoh1-CreER; Ptc^{fl/fl}* and *Atoh1-CreER; Ptc^{fl/fl}; Kif20a^{fl/fl}* knockout mice. Some tumor cells could expand in culture and we were able to establish two cell lines from each tumor type. We first examined CreER-mediated gene deletion by genotyping PCR on genomic DNAs (Supplementary Fig. 2a). Our results showed that targeted homozygous deletion of the *Ptc* gene was evident in all four tumor cell lines, consistent with the occurrence of MB in these mice (Supplementary Fig. 2b; results of one cell line from each tumor type were shown). Surprisingly, however, in the cell lines derived from the *Ptc/Kif20a* double-knockout mice, the *Kif20a* gene was in a heterozygous state, with one allele having the intact floxed *Kif20a* gene and the other allele having the floxed exons

removed (Supplementary Fig. 2c). Further quantitative PCR on RNAs isolated from these cells showed that a *Kif20a* transcript was expressed, albeit in a reduced level (Supplementary Fig. 2d). In theory, tamoxifen-induced activation of CreER should have led to simultaneous homozygous deletion of both the *Ptc* and *Kif20a* genes in GNPs carrying the *Atoh1-CreER; Ptc^{fl/fl}; Kif20a^{fl/fl}* alleles; however, our results suggested that simultaneous deletion of the *Kif20a* and *Ptc* genes was faulty in some GNPs of the double-knockout mice, such that although the *Ptc* gene was deleted, the *Kif20a* gene was left half intact. Thus, the cell lines derived from tumors of the *Ptc* single and *Ptc/Kif20a* double-knockout mice have the genotype of *Ptc^{−/−}; Kif20a^{+/+}* and *Ptc^{−/−}; Kif20a^{fl/−}*, respectively.

We next performed analyses on cell biological properties of these two types of tumor cells. Our results showed that tumor cells derived from the *Ptc/Kif20a* double-knockout mice had a slower proliferation rate (Fig. 5a), consistent with the overall longer survival time of the double-knockout mice (Fig. 3a). These tumor cells also showed fewer Ki67$^+$ proliferating cells and were prone to exiting the cell cycle (Fig. 5b), compared to tumor cells derived from the *Ptc* single-knockout mice. The two types of tumor cells displayed comparable 4N DNA contents in cell cycle analysis (Supplementary Fig. 3), suggesting that deletion of *Kif20a* in SHH-MB cells did not alter the cytokinesis status. To further test the effect of *Kif20a* deletion, we next introduced Cre enzyme (Cre-2A-GFP), or control green fluorescent protein (GFP), into the tumor cells derived from the *Ptc/Kif20a* double-knockout mice, using a lentiviral expression system. The infected cells were next labeled with EdU for 24 h before being fixed for immunostaining. Our results showed that expression of Cre in these heterozygous *Kif20a* tumor cells caused an increase of EdU$^+$Ki67$^-$ cells compared to cells expressing GFP control alone (Fig. 5c), suggesting more cells exited the cell cycle after Cre expression. In addition, cell cycle analysis of these tumor cells showed that the cells expressing Cre or control GFP displayed comparable 4N DNA contents (Fig. 5d). These results thus collectively indicated that knocking out the remaining *Kif20a* allele in the tumor cells derived from the *Ptc/Kif20a* double-knockout mice did not cause an obvious failure in cell division but led to increased exit of the cell cycle in daughter cells. Therefore, the slower growth rate of these tumor cells (Fig. 5a) might be attributed to their tendency for becoming post-mitotic cells due to the half dosage of *Kif20a* expression, which eventually leads to fewer proliferating cells in the population.

**Inducible knockdown of KIF20A in human MB cells inhibits proliferation in culture and tumor growth in xenograft.** We next examined the expression status of KIF20A in relation to human MB cells using available expression data of patient tumor samples. Our results revealed that strong KIF20A expression is positively correlated with Ki67 level in different subgroups or subtypes of MBs (Fig. 6a–c), indicating that KIF20A is positively associated with proliferation of human MB cells. To address the function of KIF20A in human MB cell proliferation, we generated

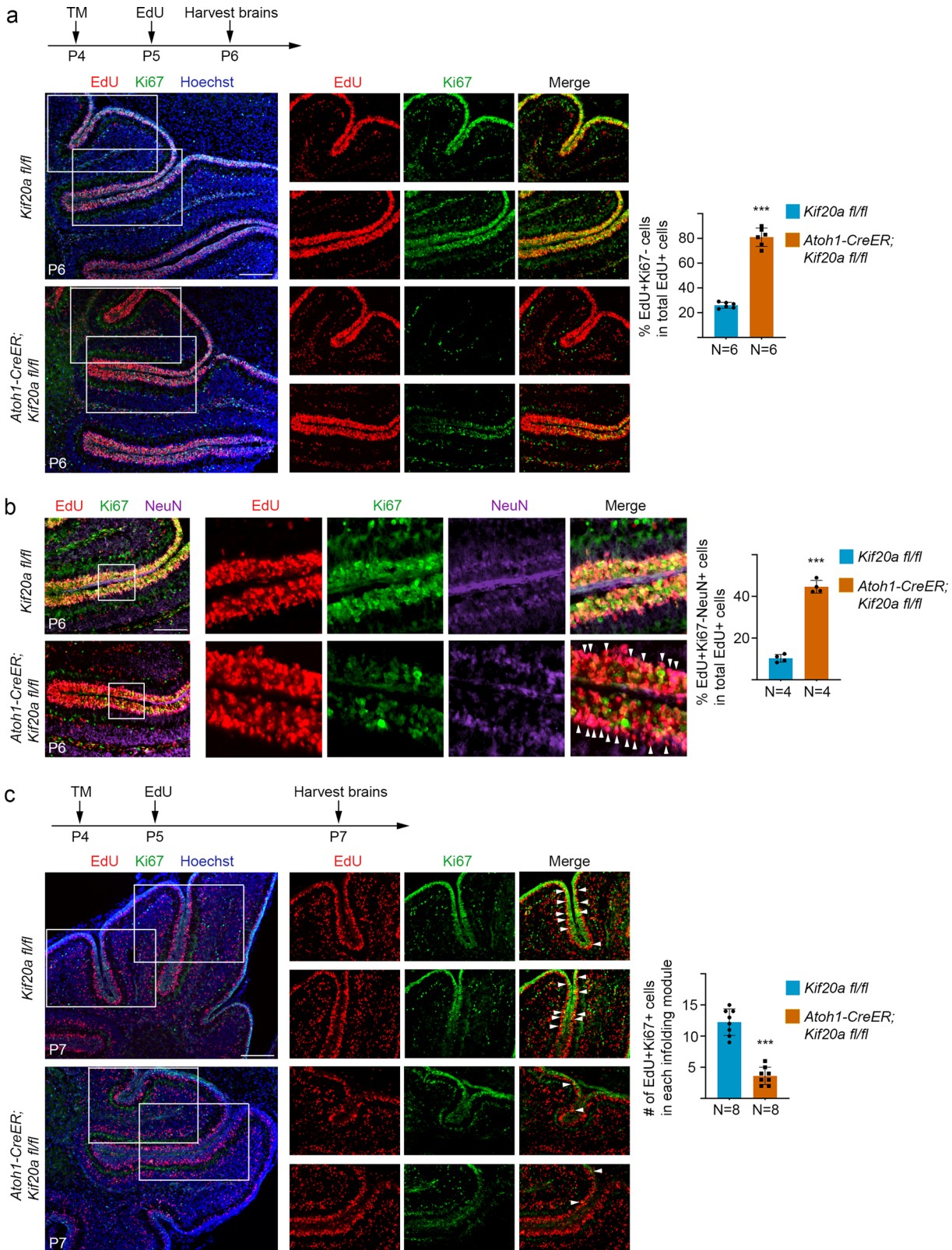

multiple shRNAs targeting human KIF20A transcript (Supplementary Fig. 4a). One such shRNA (shKIF20A726) was cloned into a Tet-on-inducible shRNA expression vector[21]. The inducible shKIF20A726 expression cassette and a GFP/firefly luciferase (ffluc) expression cassette were then stably integrated into Daoy human MB cells. Three puromycin-resistant cell clones (stably integrated with Tet-on shKIF20A726 and GFP/ffluc) were

isolated. Our data showed that doxycycline treatment could induce significant knockdown of KIF20A expression in all three cell clones (Supplementary Fig. 4b). Doxycycline induction was also seen to significantly inhibit the proliferation of all three clones (Supplementary Fig. 4c). This inhibition was not due to cytotoxicity of doxycycline, because parental Daoy cells were not affected by doxycycline treatment (Supplementary Fig. 4c).

**Fig. 2 LOF of KIF20A in GNPs causes early cell cycle exit and precocious neuronal differentiation. a** After tamoxifen (TM) treatment at P4, animal pups were labeled with EdU at P5 and then brains were collected at P6. Co-staining of EdU and Ki67 showed that knockout of *Kif20a* resulted in relatively more EdU+Ki67− cells in the EdU+ cell population compared to the wild-type littermate brains. Scale bar represents 50 µm. Data are mean ± SD. P = 1.09E − 09 (Student's *t*-test). **b** More EdU+Ki67− cells in the mutant cerebellums were positive for neuronal marker NeuN (white arrowheads). These cells formed a line outside the Ki67+ cells, reflecting they are differentiating and migrating out of the EGL. Scale bar represents 50 µm. Data are mean ± SD. P = 1.29E − 06 (Student's *t*-test). **c** After tamoxifen treatment at P4, animal pups were labeled with EdU at P5 and then brains were collected at P7. Co-staining of EdU and Ki67 showed that knockout of *Kif20a* resulted in fewer EdU+Ki67+ proliferating cells in the EGL at this stage. EdU+Ki67+ cells within individual cerebellar sulcus were used for quantification. Scale bar represents 50 µm. Data are mean ± SD. P = 1.58E − 07 (Student's *t*-test).

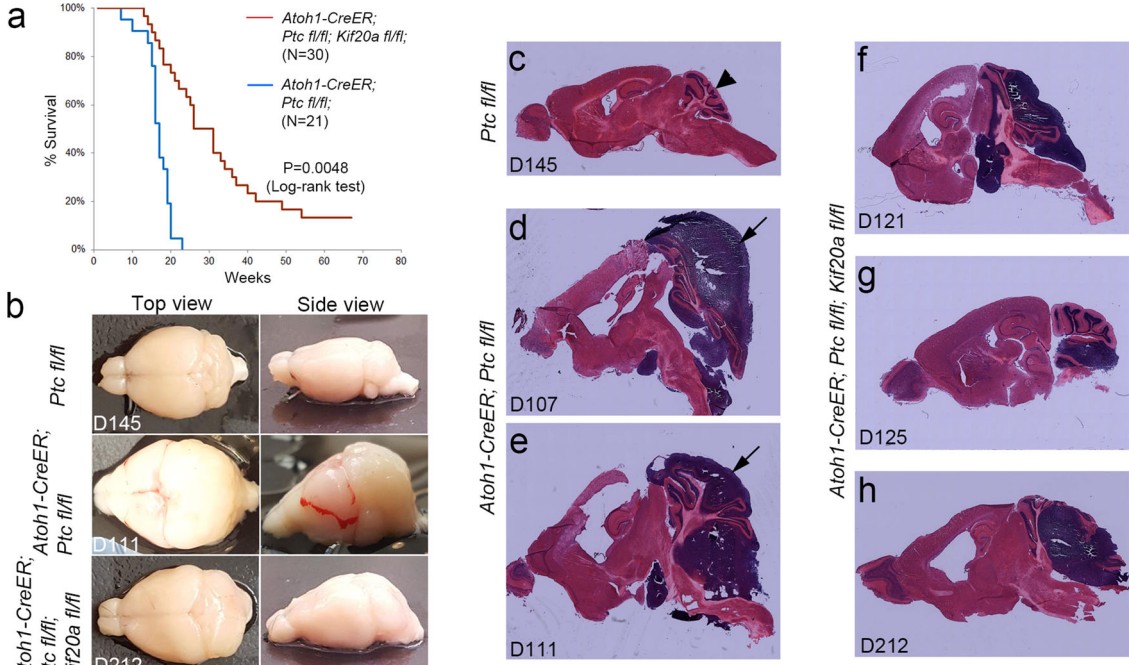

**Fig. 3 LOF of KIF20A in tumor-initiating GNPs inhibits SHH-induced MB formation. a** Single (*Atoh1-CreER; Ptc^{fl/fl}*)- and double (*Atoh1-CreER; Ptc^{fl/fl}; Kif20a^{fl/fl}*)-knockout mice were treated with tamoxifen at P4 by gavage. Brain samples were collected for analyses when brain tumor symptoms were developed. Survival of mice was summarized in the Kaplan–Meier curve. **b** Representative whole brain samples from wild-type (*Ptc^{fl/fl}*), *Ptc* single (*Atoh1-CreER; Ptc^{fl/fl}*)-, or *Ptc/Kif20a* double (*Atoh1-CreER; Ptc^{fl/fl}; Kif20a^{fl/fl}*)-knockout mice. Numbers in each panel indicate the days when brain samples were collected after TM injection (sections of these brains were shown in **c**–**h**). c Normal brain features from section of wild-type control (*Ptc^{fl/fl}*) brain collected 145 days after TM injection. Arrowhead indicates the cerebellum. **d, e** Examples of sections from *Ptc* single-knockout mice displayed strong tumor growth and often disformed brain structures. Arrows indicate the tumor mass. **f**–**h** Examples of sections from *Ptc/Kif20a* double mice displayed varied tumor sizes. The overall brain structures were in general in better shape than the single-knockout mice.

Among the three clones, Clone#2 was chosen for further in vivo studies, as it showed comparable growth rate to the parental cells (indicating minimal alteration of cell properties from lentiviral integration and/or minimal leakage of the Tet-on shKIF20A cassette).

To examine the function of KIF20A in tumor formation by Daoy cells, we injected $10^5$ Clone#2 cells into the cerebellum of each recipient NOD scid gamma (NSG) mouse using a method previously described[22]. Growth of tumor cells was then monitored by bioluminescence imaging. Four weeks after the initial cell injection, tumor growths in the brains of recipient mice were evident by the increase of imaging signal (Fig. 6d). We then separated the injected mice into two groups: one group was treated with doxycycline and the other group was untreated, as control. For the treatment group, the mice were first fed with doxycycline solution (10 mg/ml, 0.2 ml) by gavage daily for 2 consecutive days. The mice were then kept on doxycycline-containing food continuously and were monitored for tumor growth by bioluminescence imaging. Our results revealed that, in comparison to the control group, the group of mice that received treatment showed significantly slower growth of brain tumors

(Fig. 6d) and better overall survival (Fig. 6e), indicating that inhibition of KIF20A expression could suppress cell proliferation in a growing tumor.

## Discussion

In mouse spontaneous brain tumor models, accumulating evidence from genetic studies of the MB[20,23–25] or glioma[26–30] have demonstrated that brain tumors originate from dysregulated stem or progenitor cells. Gene expression profiling analyses have also revealed that brain tumor cell transcriptomes display molecular characteristics resembling neural stem/progenitor cell types[31–37]. These data thus suggest that targeting the proliferation or differentiation programs employed by stem/progenitor cells is expected to effectively interfere with brain tumor initiation or growth. Along these lines, the daughter cells' fate specification process during stem/progenitor cell divisions presents a particularly good opportunity for therapeutic intervention, because the process is thought to occur during the late stages of mitosis and is expected to lie downstream of many proliferation-promoting (oncogenic) signaling pathways. In this report, our genetic analyses of KIF20A in both normal and cancer-initiating cerebellar

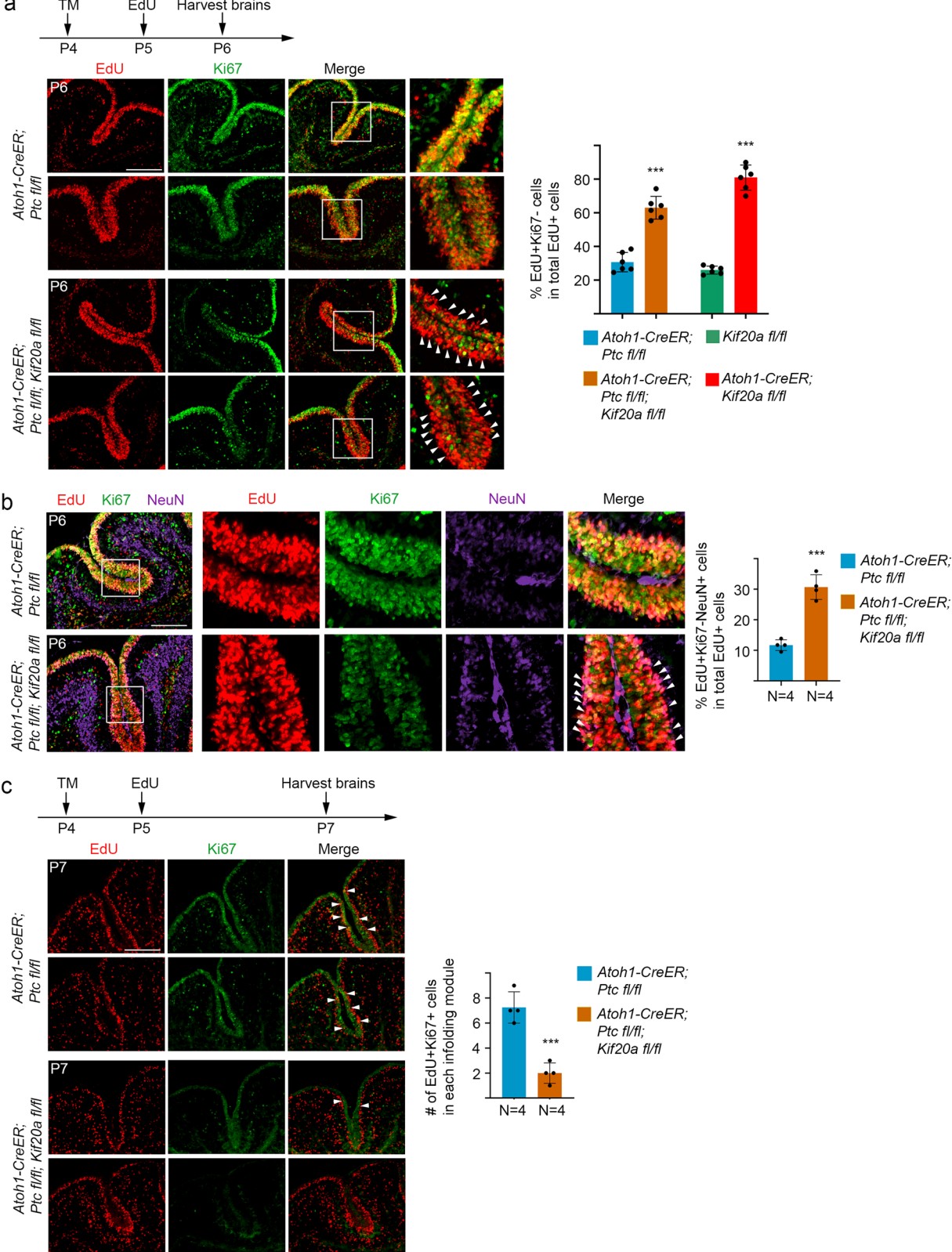

GNPs presents an experimental validation of a cell division mode regulator as a potential target for brain tumor inhibition.

One interesting observation from our genetic analyses was that MB cells isolated from the *Atoh1-CreER; Ptc^{fl/fl}; Kif20a^{fl/fl}* double-knockout mice were homozygous null for the *Ptc* gene but heterozygous for the *Kif20a* gene. This result suggested that in the *Atoh1-CreER; Ptc^{fl/fl}; Kif20a^{fl/fl}* knockout mice, some GNPs responded to tamoxifen induction with one *Kif20a* allele being deleted but leaving the other *Kif20a* allele intact. One possible reason for this phenomenon might be that Cre recombinase had lower recombination efficiency at the *Kif20a* locus than at the *Ptc* locus. This could be due to some structural hindrance of the *Kif20a* gene; for instance, ten exons were floxed in the *Kif20a* gene but only one exon was floxed in the *Ptc* gene

**Fig. 4 Inducible knockout of *Kif20a* in tumor-initiating GNPs causes early cell cycle exit. a** After tamoxifen treatment at P4, animal pups were labeled with EdU at P5 and then brains were collected at P6. Co-staining of EdU and Ki67 revealed that knockout of *Kif20a* in MB-initiating GNPs (*Ptc* single-knockout GNPs) resulted in more EdU$^+$Ki67$^-$ cells in the EdU$^+$ cell population. White arrowheads indicate the EdU$^+$Ki67$^-$ cells, many of which line outside the proliferating cell zone. Scale bar represents 50 μm. Quantifications from Fig. 2a (green and red columns) were plotted together in the graph. Data are mean ± SD. $P = 1.82E-05$ (between the first two columns) (Student's *t*-test). **b** More EdU$^+$Ki67$^-$ cells in the *Ptc/Kif20a* double-knockout cerebellums were positive for neuronal marker NeuN (white arrowheads). Scale bar represents 50 μm. Data are mean ± SD. $P = 0.00013$ (Student's *t*-test). **c** After tamoxifen treatment at P4, animal pups were labeled with EdU at P5 and then brains were collected at P7. Co-staining of EdU and Ki67 showed that knockout of *Kif20a* in MB-initiating GNPs resulted in fewer EdU$^+$Ki67$^+$ proliferating cells in the EGL. EdU$^+$Ki67$^+$ cells within individual cerebellar sulcus were used for quantification. Scale bar represents 50 μm. Data are mean ± SD. $P = 0.00042$ (Student's *t*-test).

(Supplementary Fig. 2a). Another reason might be that Cre-mediated recombination could generate mosaic patterns of recombination products between the *Ptc* and *Kif20a* genes, such as cells with genotypes of *Ptc$^{-/-}$; Kif20a$^{fl/-}$* or *Ptc$^{fl/-}$; Kif20a$^{-/-}$*, or other combinations. The GNPs of the *Ptc$^{-/-}$; Kif20a$^{fl/-}$* genotype were detected in tumor cells, because they should have had an advantage in proliferation over GNPs of other combinations of genotypes and, as a result, they eventually grew out into a tumor over the time. Regardless of how the *Ptc$^{-/-}$; Kif20a$^{fl/-}$* GNPs were produced, our results would indicate that the prolonged survival time in the tamoxifen-induced *Atoh1-CreER; Ptc$^{fl/fl}$; Kif20a$^{fl/fl}$* mice (Fig. 3a) might be underestimated, due to the incomplete deletion of the *Kif20a* gene in some of the cancer-initiating progenitor cells. In support of this thought, further deletion of the remaining allele of the *Kif20a* gene in tumor cells derived from the *Ptc/Kif20a* double-knockout mice (cells having a genotype of *Ptc$^{-/-}$; Kif20a$^{fl/-}$*) accelerated cell cycle exit (Fig. 5c).

Our data obtained from *Atoh1-CreER; Kif20a$^{fl/fl}$* single and *Atoh1-CreER; Ptc$^{fl/fl}$; Kif20a$^{fl/fl}$* double-knockout mice together showed that KIF20A functions similarly in normal and cancer-initiating GNPs to maintain their proliferative potential. We previously found that blocking KIF20A function in NPCs of the developing cerebral cortex did not compromise the divisions of parental progenitor cells but drive the daughter cells to adopt a differentiative fate[9]. Similarly, data from this study indicated that GNPs with LOF of KIF20A were able to complete cell divisions but the nascent daughter cells became differentiated into neurons, leading to early depletion of proliferating cells in the developing cerebellum and in MB. These results thus indicate that targeting KIF20A, or other regulators of cell division mode control in a broad sense, is distinct from conventional anti-mitotic inhibitors, which act primarily to induce mitotic arrest or cytokinesis defect. Blocking the function of a cell division mode regulator does not disrupt a progenitor cell's division process per se but promotes the daughter cells to take a differentiative path. Targeting the process of daughter cell-fate specification may thus present a new strategy for expanding the repertoire of anti-proliferation chemotherapy for malignant brain tumors.

## Methods

**Antibodies**. The following antibodies were used: KIF20A(L-13) (Santa Cruz, sc-104954), KIF20A (OriGene, AP01361PU-N), Alpha-tubulin(TU-01) (Thermo-Fisher, MA1-19162), Ki67(SP6) (Abcam, Ab16667), Active caspase 3 (BD Biosciences, 559565), and NeuN (Millipore, MAB377). Secondary antibodies were purchased from Jackson ImmunoResearch Laboratories (Cy3, Cy5, and Cy2 AffiniPure conjugated) and Click-iT®EdU Alexa Fluor®594 image kit (ThermoFisher Scientific) was used.

**DNA constructs and lentivirus production**. Lentiviral transfer plasmids used in this study include the following: Hiv7CMV-Cre-myc-2A-GFP (Addgene #117148), Hiv7CMV-GFP, Hiv7CMV-GFP-IRES-Luciferase, and Tet-on-inducible shRNA plasmids Tet-pLKO-shKIF20A726 or Tet-pLKO-shKIF20A181-Puro (shRNA sequences were cloned into pLKO-Tet-On plasmid, Addgene # 21915 at AgeI/EcoRI sites). The plasmids used for shRNA Luciferase in vitro screening include the following: pNUTS-shScramble[9], pNUTS-shKIF20A726 or pNUTS-shKIF20A181, and shRNA template plasmid pSiCheck2.2-hKIF20A-CDS.

Lentiviruses were generated by co-transfecting 293T cells with 15 μg Lentiviral transfer plasmids, 15 μg second-generation lentiviral packaging plasmid psPAX2 (Addgene #12260), and 5 μg lentiviral envelope plasmid pCMV-G in 10 cm tissue culture plate using calcium phosphate cell transfection reagents. Growth media was exchanged around 6 h later and lentivirus-containing supernatant was collected 48 h later.

**Mice**. *Kif20a$^{fl/fl}$* mice were deposited in Mutant Mouse Resources and Research Centers (MMRRC) (strain ID: MMRRC_050513-UCD). *Atoh1-CreER; Ptc$^{fl/fl}$* mice were obtained from Dr. Robert Wechsler-Reya's group. The two strains were crossed to generate *Atoh1-CreER; Kif20a$^{fl/fl}$* and *Atoh1-CreER; Ptc$^{fl/fl}$; Kif20a$^{fl/fl}$* mice. For analyses of normal and MB-initiating GNPs in knockout animals, all mice were treated with a single dose of tamoxifen by gavage at P4, as described in the SHH-MB model[20]. Animals were group housed and maintained in the temperature range and environmental conditions recommended by Association for Assessment and Accreditation of Laboratory Animal Care (AAALAC). Animal procedures were approved by the Institutional Animal Care and Use Committee of Beckman Research Institute of the City of Hope and were carried out in accordance with NIH guideline and the Guide for the Care and Use of Laboratory Animals.

**shRNA design and screening**. Potential shRNA sequences were selected using web-based design tool (https://rnaidesigner.thermofisher.com/rnaiexpress/design.do). shRNAs were expressed under the control of a mouse U6 promoter in pNUTs vector, which additionally contains a ubiquitin promoter-EGFP expression cassette. Candidate shRNAs in pNUTs and cDNA target in psi-CHECK were co-transfected into HEK293 cells in triplicates; 48 h later, the firefly and *Renilla* Luciferase values were determined with Promega's Dual-Luciferase® Reporter Assay System (Promega, E1910). The final inhibition unit (% luciferase activity) was the normalized value (*Renilla*/Firefly). The sequences of 19mer human KIF20A shRNAs (the loop of shRNA is underlined) are as below:

Scrambled shRNA, 5′-CGGCTGAAACAAGAGTTGG-<u>TTCAAGAGA</u>-CCAA CTCTTGTTTCAGCCG-3′;

shKIF20A726, 5′-GAGGAGTGTCTACATCGAA-<u>TTCAAGAGA</u>-TTCGATGT AGACACTCCTC-3′;

shKIF20A181, 5′-AGTATGGAGAAGGTGAAAG-<u>TTCAAGAGA</u>-CTTTCAC CTTCTCCATACT-3′.

**Western blotting**. Daoy cell clones stably integrated with Tet-on shKIF20A726 were cultured with or without Doxycycline induction for 3 days. These Daoy cells were washed with phosphate-buffered saline (PBS), lysed with 2× SDS loading buffer, and boiled for 5 min. Denatured proteins were resolved by SDS-polyacrylamide gel electrophoresis and transferred to a polyvinylidene difluoride membrane for western blotting detection by KIF20A (L13) antibody (1 : 500) and horseradish-conjugated donkey anti-goat secondary antibody (1 : 2500) with chemiluminescent substrate (ThermoFisher Scientific, Cat: 34095).

**Histology, immunocytochemistry, and immunohistochemistry**. Animal were perfused with 1× PBS followed by 4% paraformaldehyde (PFA). The whole brains were removed, fixed in 4% PFA overnight, cryoprotected in 30% sucrose, and embedded in Tissue Tek OCT Compound (Fisher HealthCare). For histological analysis, whole brain sagittal sections (6 μm) were stained with hematoxylin and eosin (Sigma). For immunohistochemistry, whole brain sagittal sections (16 μm) or fixed tumor cells were blocked and permeabilized for 2 h at room temperature in blocking solution (1× PBS, 0.1% Triton X-100, 10% Donkey serum, 5% AffiniPure Fab fragment Donkey anti-Mouse IgG, 0.2% Sodium Azide), followed by incubation with primary antibody in blocking solution at 4 °C overnight. The sections or cells were rinsed three times with 1× PBS and incubated with secondary antibody in 1× PBS for 1 h at room temperature. Sections or cells were counterstained with Hoechst 33342 dye and rinsed three times with 1× PBS followed by once in water, then the sections or cells were mounted with Fluoromount-G (SouthernBiotech) for imaging with microscope.

**Analyses of inducible knockout mice**. The pups of *Atoh1-CreER; Ptc$^{fl/fl}$* and *Atoh1-CreER; Ptc$^{fl/fl}$; Kif20a$^{fl/fl}$* mice were treated with tamoxifen at P4 by oral

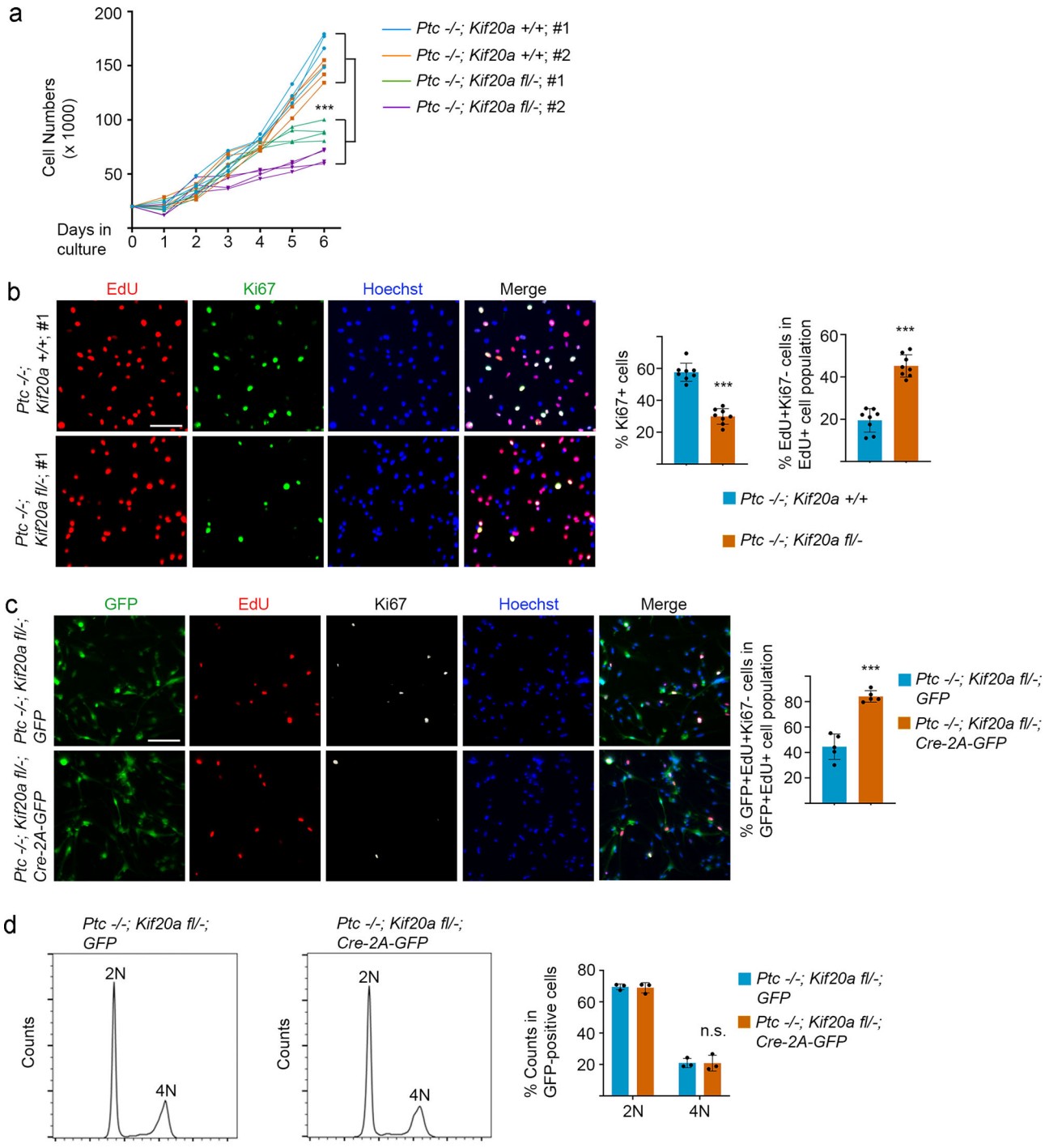

**Fig. 5 LOF of KIF20A in mouse tumor cells inhibits proliferation by inducing cell cycle exit. a** Two tumor cell lines were derived from the *Ptc* single-knockout mice and these cells carried a genotype of $Ptc^{-/-}; Kif20a^{+/+}$. Two tumor cell lines from the *Ptc/Kif20a* double-knockout mice were also established in culture and these cells showed a genotype of $Ptc^{-/-}; Kif20a^{fl/-}$. The latter two tumor lines displayed slower proliferation rates. Individual data points represented replicates of cell samples. Data are mean ± SD. $P = 9.46E - 05$ (two-way ANOVA). **b** Proliferating tumor cells were labeled with EdU for 24 h and were then stained for EdU and Ki67. There were fewer Ki67+ cells and a relatively high percentage of EdU+Ki67− cells (in the total population of EdU+ cells) in the tumor line derived from the *Ptc/Kif20a* double-knockout mice. Scale bar represents 50 μm. Data are mean ± SD. $P = 5.45E - 08$; $1.75E - 07$ (Student's t-test). **c** Tumor cells (having a genotype of $Ptc^{-/-}; Kif20a^{fl/-}$) derived from the *Ptc/Kif20a* double-knockout mice were infected with lentivirus expressing Cre-2A-GFP or control GFP and were next labeled with EdU for 24 h in culture. Knockout of the remaining *Kif20a* allele in these cells resulted in cell cycle exit. Scale bar represents 50 μm. Data are mean ± SD. $P = 4.12E - 05$ (Student's t-test). **d** Tumor cells derived from the *Ptc/Kif20a* double-knockout mice were infected with lentivirus expressing Cre-2A-GFP or control GFP. After labeling with propidium iodide, the cells were examined for their DNA contents by flow cytometry analysis. PI, propidium iodide; 2N, cells in G1 phase; 4N, cells in G2/M phase or bi-nucleated cells. n.s., not significant.

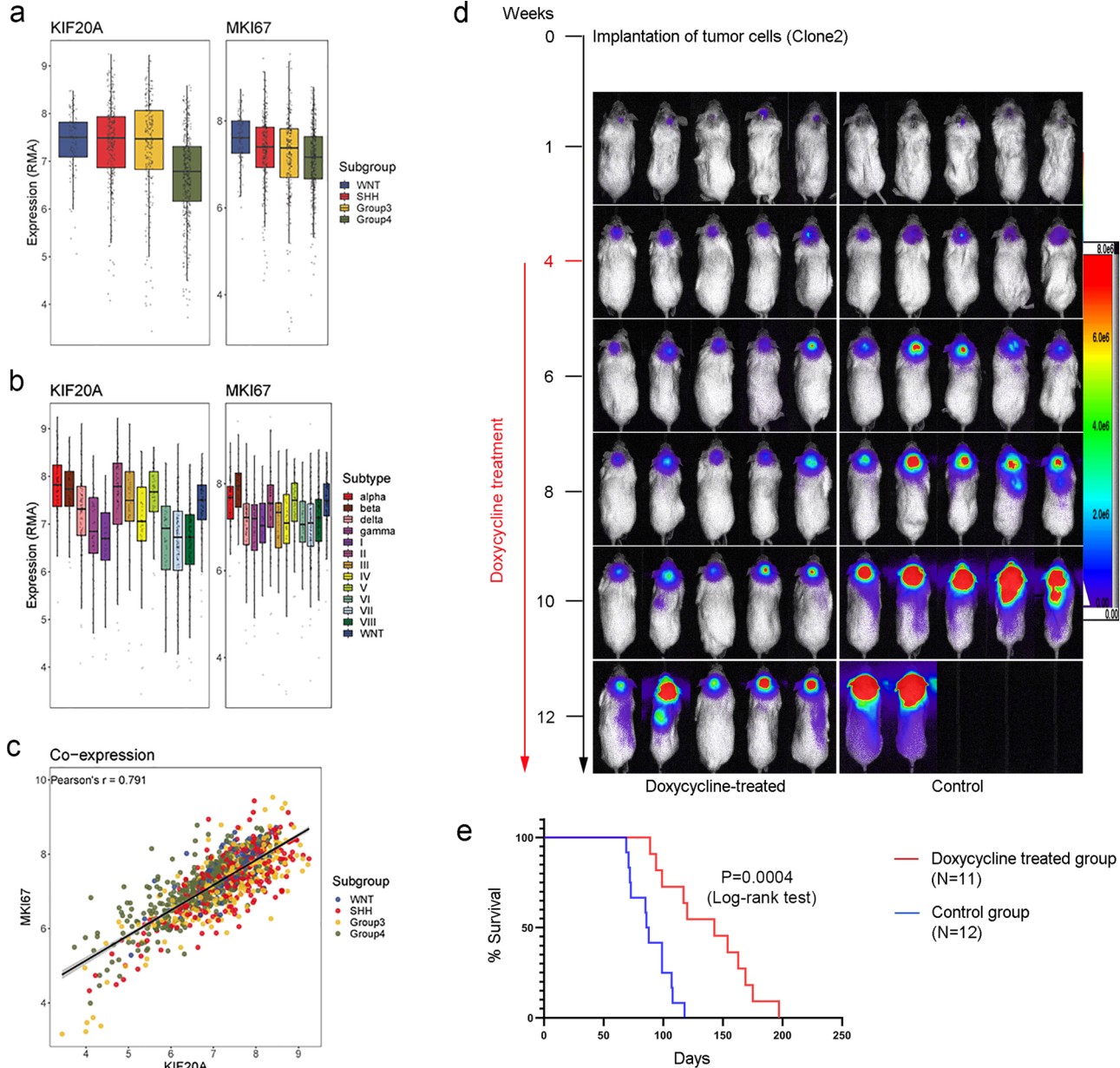

**Fig. 6 Knockdown of KIF20A expression in human MB cells inhibits tumor growth. a** Different subgroups of MB patient cells display strong expression of KIF20A, which is positively correlated with proliferating cell marker Ki67. **b** KIF20A and Ki67 expressions are also positively correlated in different subtypes of human MB cells. **c** Comparison between KIF20A expression and Ki67 expression in subgroups of human MB cells. Expression of the two factors are highly correlated. **d** Daoy cells stably integrated with Tet-on shKIF20A726 and firefly luciferase were intracranially injected into the cerebellar areas of recipient NSG mice ($10^5$ cells per mouse). Tumor growths were monitored by bioluminescence imaging. Doxycycline treatment started at 4 weeks after the initial cell implantation (indicated with a red font and line). The treatment group of mice was first given doxycycline by gavage for 2 consecutive days and was then fed with doxycycline-containing food for the entire duration of the experiment. **e** Survival data were analyzed by Kaplan–Meier plot.

gavage (0.6 mg/30 µl) using 24 G gavaging needles. Tamoxifen (T5648, Sigma) was prepared as a 20 mg/ml stock solution in corn oil (Sigma). EdU was then given by intraperitoneal injection at P5 with the dosage of 0.1 mg per gram of the pups. After 24 or 48 h post EdU treatment, the brain samples were collected at P6 and P7 for analyses. Cryosections of P6 or P7 brains (16 µm) were analyzed for immunostaining on EdU, Ki67 (1 : 200), NeuN (1 : 100), or activated caspase 3 (1 : 200). Stained sections were mounted with Fluoromount-G for imaging. For MB formation analysis, after pups were treated with tamoxifen at P4 by gavage, they were monitored until symptoms of brain tumor growth occurred. Brain tumor samples were removed and cryosections were made for histological documentation.

For fluorescence-activated cell sorting (FACS)-based cell cycle analysis, pieces of P6 cerebellums were gently dissociated in cold Hank's balanced salt solution (HBSS) (Mediatech) with 5 mM EDTA. After centrifugation with $300 \times g$ at 4 °C for 10 min, cells were washed twice with cold HBSS buffer. Subsequently, large cell/ aggregates were removed with a cell strainer (40 µm). After fixation with 70%

ethanol drop by drop and then being stored for overnight at 4 °C, cells were washed in cold PBS twice. For propidium iodide staining, cells were resuspended in 0.5 ml staining buffer (PBS + 10 µg/ml PI + 0.1% Triton X-100 + 0.2 mg/ml DNAse-free RNAse A) and were incubated at 37 °C for 30 min. Cells were filtered before FACS analysis. Measurements were done using a CyAn ADP instrument (DakoCytomation) in combination with summit software.

**Expansion of mouse MB cells in culture.** Brains with MB were dissected out and transferred into PBS on ice. Blood and fat tissues were removed from the tumor mass under a dissecting microscope and the tumor tissue pieces were washed again with PBS. Tumor tissues were minced with a scalpel, passed through syringes with 18- and 22-gauge needles, and then incubated in a 1 : 1 mixture of Accutase (Sigma) and TrypLE (Invitrogen) for 15 min at 37 °C with occasionally vortex. Undigested tissues were removed by filtering with 40 µm cell strainer (Falcon,

352340). Dissociated cells were washed twice with Dulbecco's modified Eagle medium (DMEM)/F12 medium followed by centrifugation at 2000 r.p.m. for 5 min. Cells were then resuspended and plated onto uncoated dishes in Neurobasal and DMEM/F12 media (1 : 1 mix) containing N2 and B27 supplements (Invitrogen), and human recombinant FGF2 and EGF (20 ng/ml, PEPROTECH). Five to 7 days later, cell spheres were dissociated in Accutase (Sigma-Aldrich) and plated onto Primaria dishes (BD Biosciences) coated with Poly-L-ornithine solution (Sigma-Aldrich) and mouse laminin (Sigma-Aldrich) to allow adherent growth.

**Generation of Daoy cell lines with stably integrated Tet-on-inducible shRNA expression system**. Daoy cells were infected with lentivirus of Tet-on-inducible human KIF20A shRNA-726 and GFP-Luciferase prepared from Tet-pLKO-shKIF20A726 plasmid and Hiv7CMV-GFP-IRES-Luciferase plasmid, respectively. The transduced Daoy cells were selected for stable single clones with Puromycin (1 µg/ml, Sigma-Aldrich) and GFP fluorescence signal in DMEM (Invitrogen) with 10% fetal bovine serum. Single clones were picked up by glass cloning cylinders (Sigma-Aldrich) in 10 cm plate. Three clones were obtained and tested for growth ability and doxycycline (0.5 µg/ml) inducibility using growth curve assay, western blottings, and luciferase reporter assay system. Clone #2 was selected for further use.

**Growth assay of tumor cells**. Mouse MB cell lines or Daoy clones stably integrated with Tet-on-inducible human KIF20A shRNA-726, or wild-type Daoy cells were seeded into multiple wells of 24-well plates ($2 \times 10^4$ or $1 \times 10^4$ cells per well). Cells in quadruplicate were then dissociated and counted using a hemocytometer at consecutive days after initial plating.

**Analyses of KIF20A expression using transcriptome databases of MB patient samples**. Raw microarray CEL files were obtained from the two published MB cohorts, GSE85217[38] and EGAS00001001953[39]. Expression data were quantified using custom chip definition files corresponding to Ensembl genes (v23.0.0) from brain array[40] and then each dataset was normalized using robust multichip averaging from the oligo package v1.54.1[41]. Duplicate samples were removed and then datasets were integrated for batch correction using ComBat from the sva package v3.36.0[42].

**Intracranial tumor transplantation and bioluminescence imaging**. NSG mice (8–10 weeks old) were anesthetized with isoflurane and oxygen, and were placed under a microscope. After exposing the skull with a scalpel, a cell suspension ($1 \times 10^5$ cells in 4 µl PBS) of Daoy clone #2 stably integrated with Tet-on-inducible human KIF20A shRNA (shRNA-726) and Luciferase gene was slowly injected into the cerebellum at a depth of 2.5 mm using a 10 µl Hamilton syringe with a 26 G needle, using a plastic blocker to control the injection depth. After injection, the incision was closed using wound clips. The transplanted mice were then monitored weekly with luciferase-based bioluminescence imaging for tumor growth. When tumor growth was evident (4 weeks after cell implantation), the mice were separated into two groups—control group and doxycycline treatment group. For the treatment group, mice were first gavaged with 0.2 ml of Doxycycline (10 mg/ml) for 2 consecutive days. Then the mice were fed continuously with Doxycycline-containing food (TestDiet, 625 p.p.m.), with the food being replaced every other day. Tumor growths in two mice groups were monitored weekly by bioluminescence imaging. Survivals of mice after cell transplantation were recorded and analyzed.

**Image acquisition and processing**. Fluorescent images were taken with Zeiss Observer II or confocal microscope of Zeiss LSM 700 or Zeiss LSM 880.

**Statistics and reproducibility**. The numbers of tumor-forming knockout or NSG mice (animal survival experiment), brain samples used in each genetic analysis, or replicates of cell samples (wells) in cultured cell experiments, were indicated in the histograms of the figures. All data were presented as mean ± SD. Student's t-test or two-way analysis of variance were performed and indicated in histograms of the figures. Log-rank test was performed for the Kaplan–Meier curve. In all descriptions, $*p < 0.05$, $**p < 0.01$, and $***p < 0.001$; n.s. represented not significant ($p > 0.05$). Cultured cell-related experiments were repeated twice and similar results were obtained.

**Reporting summary**. Further information on research design is available in the Nature Research Reporting Summary linked to this article.

## Data availability

All data and materials are available from the authors upon reasonable request. The sources of most materials were indicated in the "Methods" section. All source data underlying the graphs and charts presented in the main figures are available as Supplementary Data.

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

## Acknowledgements
We thank Jeremy LaDou and animal center staff for assistance with animal breeding and care, and Brian Armstrong and staff for the use of imaging facility. This work is supported by NIH grant NS096130 to Q.L.

## Author contributions
R.X.Q. and Q.L. conceived and designed the study. R.X.Q. and J.W. conducted the experiments. B.G. and P.A.N. provided analyses of KIF20A expression in medulloblastoma patient transcriptome databases. R.W.R. provided *Atoh1-CreER; Ptc^{fl/fl}* mice and insightful discussions on medulloblastoma biology. R.X.Q. and Q.L. analyzed data and wrote the manuscript.

## Competing interests
The authors declare no competing interests.
