## [Peer Review File · Communications Biology]

Reviewers' comments:

Reviewer #1 (Remarks to the Author):

In this manuscript, Qiu et al builds upon a series of previous work from their lab on KIF20A and its important role in balancing symmetric versus asymmetric divisions in cortical NPCs during development. In the current work, the authors tested the exciting possibility that the balance between symmetric and asymmetric divisions can be manipulated as a potential therapeutic target for treating brain tumors. The authors used genetic analyses to show that Kif20a is required for proliferation of both normal and cancerous cerebellar GNPs, and used an elegant transplantation paradigm wherein Daoy human MB cells, with or without KIF20A manipulation, were intracranially injected into the cerebellum of recipient mice. The survival data for both the genetic models and the transplantation model are quite convincing. The observation that cells that escaped deletion of both Kif20a alleles were present in tumors is very interesting and strengthens the authors' conclusions. Overall, this work is well done and has potential to inform a novel therapeutic approach that is distinct from conventional anti-mitotic inhibitors. I only have a few minor experimental and textual comments.

1. For the Atoh1-CreER, the authors used a single Tamoxifen injection on P4. Can the authors provide data on the efficiency and consistency of Kif20a deletion? This can be done, for example, by KIF20A immunostaining, which looks very nice in Figure 1. In particular, is KIF20A gone one day after tamoxifen, on P5, when EdU was injected?
2. Overall, the quantifications were carefully done. The authors should however show or overlay individual data points on the bar graphs.
3. The cell cycle exit/re-entry data in Figure 4 should be plotted together with data from Figure 2 for side by side comparison.
4. Like all dividing cells, NPCs have multiple anti-tumorigenic mechanisms (e.g. p53, cell cycle checkpoints, etc.). I suggest adjusting the language in the intro that seems to suggest symmetric/asymmetric divisions to be the only mechanism for this in NPCs.
5. If the Kif20a/Ptc dKO tumor cells showed increased cell cycle exit, shouldn't this be reflected in the flow cytometry-based cell cycle analysis in Figure 5?

Reviewer #2 (Remarks to the Author):

Qiu et al show how SHH medulloblastoma can be suppressed by targeting KIF20A and cell division in cerebellar progenitor cells. The authors use conditional knock outs of Kif20a, cell cycle analysis in vivo and study Math1-driven Ptch loss during brain tumor development. They further show that KIF20A suppression by shRNA can inhibit proliferation also in human SHH medulloblastoma. Still, the report does not demonstrate any clear role of KIF20A in medulloblastoma development. Although the paper is well written and most experiments are carefully performed, I have a few concerns and suggestions for improving the manuscript.

Major:

1. Authors never motivate why picking P7 as the time point for checking mitosis. This is likely when cell division is peaking. However, why did authors not investigate if cell cycling of Kif20a depleted GNPs ends at a postnatal time point that is earlier than that of control animals? This data would be informative. Differentiation/migration at a later time point than P7 (like P21) would also be informative in order to understand this potential neuronal differentiation process.
2. The authors claim that by depletion of Kif20a the cells differentiate to become more mature neurons rather than going into apoptosis or cell cycle arrest. Still, there is no proof of this differentiation in Fig. 2. For example, in Fig 2b when showing NeuN staining, the authors have not quantified this or included NeuN staining of any control animals with wild-type Kif20a. Authors

show NeuN staining, likely negative, in Fig 1a, which is not commented on in the results. This must be an important difference to quantify in order to understand the differentiation process. Further, Cl. Casp 3 staining of Kif20a depletion and a cell cycle analysis would be useful in order to rule out apoptosis and confirm arrest as compared to controls.

3. The authors claim that depletion of Kif20a also in MB promotes differentiation of nascent progenitors. Still, again one would expect an increase in the number of differentiated cells in these animals, like an increased number of NeuN positive cells, or similar.

4. Is increased KIF20A a marker of poor prognosis or at least elevated in SHH MB? Further is it correlating with Ki67 levels in such tumors? There is plenty of published expression data from patients available to understand and at least report this data here.

5. The mechanism of Kif20a as crucial in balancing symmetric versus asymmetric cell division in these cells is referred to and claimed in both introduction and discussion but never really examined in the paper. Why did authors not show any proof of this using markers after division (e.g. NUMB, TRIM3) or checking correlation with the previously studied SEPT7 protein or similar.

Minor

1. Tumors escape Kif20a depletion as these mice still die from large tumors. Is there any difference in cytokinesis or Kif20a-dependent intracellular transportation in these tumors as compared to Kif20a wild type tumors? Would the cytokinesis function of Kif20a in these tumor cells be compensated to a large extent by the homologous protein Kif20b or by other mechanisms?

2. The authors suggest that SHH MB could be potentially targeted by a drug inhibiting Kif20a. Still it is not sure if any pathway in or downstream of the SHH pathway is affected by this depletion? This could then perhaps be coupled to early differentiation of GNPs. Even if such a SHH marker might be difficult to test in DAOY cells that are not considered to faithfully mimic SHH MB cells, I guess it would be easier and more relevant to find this in the presented mouse model where at least downstream Mycn levels would likely decrease.

Responses to Reviewers' Comments

Reviewer #1 (Remarks to the Author):

In this manuscript, Qiu et al builds upon a series of previous work from their lab on KIF20A and its important role in balancing symmetric versus asymmetric divisions in cortical NPCs during development. In the current work, the authors tested the exciting possibility that the balance between symmetric and asymmetric divisions can be manipulated as a potential therapeutic target for treating brain tumors. The authors used genetic analyses to show that Kif20a is required for proliferation of both normal and cancerous cerebellar GNP, and used an elegant transplantation paradigm wherein Daoy human MB cells, with or without KIF20A manipulation, were intracranially injected into the cerebellum of recipient mice. The survival data for both the genetic models and the transplantation model are quite convincing. The observation that cells that escaped deletion of both Kif20a alleles were present in tumors is very interesting and strengthens the authors' conclusions. Overall, this work is well done and has potential to inform a novel therapeutic approach that is distinct from conventional anti-mitotic inhibitors. I only have a few minor experimental and textual comments.

1. For the Atoh1-CreER, the authors used a single Tamoxifen injection on P4. Can the authors provide data on the efficiency and consistency of Kif20a deletion? This can be done, for example, by KIF20A immunostaining, which looks very nice in Figure 1. In particular, is KIF20A gone one day after tamoxifen, on P5, when EdU was injected?

Thank you for the positive comments.

The single tamoxifen injection at P4 was adopted from the SHH-induced spontaneous medulloblastoma model, in order to ensure consistency between our analyses of normal and cancerous GNPs. A description of this protocol was now included in the Method section (in red font, page 14).

To examine the efficiency of tamoxifen-induced Kif20a deletion, we tested several other antibodies (the original L-13 anti-KIF20A antibody used in our earlier experiments was discontinued from Santa Cruz Biotechnology), including monoclonal D-3 anti-KIF20A from Santa Cruz, rabbit polyclonal anti-KIF20A from OriGene, rabbit polyclonal anti-KIF20A from Biorbyt. Only OriGene antibody could work on frozen brain tissues and the data obtained from P5 and P6 Kif20a knockout and control brains were shown in Supplementary Fig. 1a. The results indicated that Kif20a deletion occurred within 24 hours after tamoxifen induction, which was consistent with our new data of Ki67 staining on P5 brains (Fig. 1b), showing noticeable differences of proliferating cells between the Kif20a knockout and control. The information for OriGene anti-KIF20A antibody was included in the Method (page 13).

2. Overall, the quantifications were carefully done. The authors should however show or overlay individual data points on the bar graphs.

We have revised the bar graphs with overlay data points in all genetic data Figures.

3. The cell cycle exit/re-entry data in Figure 4 should be plotted together with data from Figure 2 for side by side comparison.

We have plotted Fig. 2a data together with Fig 4a data in the same quantification graph (revised Fig. 4a).

4. Like all dividing cells, NPCs have multiple anti-tumorigenic mechanisms (e.g. p53, cell cycle checkpoints, etc.). I suggest adjusting the language in the intro that seems to suggest symmetric/asymmetric divisions to be the only mechanism for this in NPCs.

We have revised the wordings accordingly (page 3).

5. If the Kif20a/Ptc dKO tumor cells showed increased cell cycle exit, shouldn't this be reflected in the flow cytometry-based cell cycle analysis in Figure 5?

There was a subtle decrease of S phase cells in Kif20a/Ptc dKO tumor cell population in flow cytometry-based cell cycle analysis, but the overall FACS profiles did not show significant differences between the two cell types. A subtle reduction of S phase cells but retaining the overall FACS profiles were similarly observed in our new analyses of primary cells derived from Kif20a knockout and control cerebellums (Fig. 1d) or in our previous analyses of Kif20a knockout and control cortical NPCs (Geng et al., Nature Comms. 2018).

The reason for this subtle difference in FACS-based cell cycle analysis was not very clear. Perhaps this assay was a snapshot of cell cycle status at the time point of cell fixation, whereas the cell cycle exit analysis provided a chase after BrdU or EdU labeling to monitor the evolving behaviors of the cells (different progenitor cells leaving or re-entering cell cycle within the 24 hour period of chase), so the latter method might better reveal the cell cycle exit or re-entry dynamics in a developing brain system. In our studies of KIF20A or SEPT7 function, the FACS-based cell cycle analysis did provide a more conclusive assay to quantitatively assess potential defects of cytokinesis.

Reviewer #2 (Remarks to the Author):

Qiu et al show how SHH medulloblastoma can be suppressed by targeting KIF20A and cell division in cerebellar progenitor cells. The authors use conditional knock outs of Kif20a, cell cycle analysis in vivo and study Math1-driven Ptch loss during brain tumor development. They further show that KIF20A suppression by shRNA can inhibit proliferation also in human SHH medulloblastoma. Still, the report does not demonstrate any clear role of KIF20A in medulloblastoma development. Although the paper is well written and most experiments are carefully performed, I have a few concerns and suggestions for improving the manuscript.

Major:

1. Authors never motivate why picking P7 as the time point for checking mitosis. This is likely when cell division is peaking. However, why did authors not investigate if cell cycling of Kif20a depleted GNP ends at a postnatal time point that is earlier than that of control animals? This data would be informative. Differentiation/migration at a later time point than P7 (like P21) would also be informative in order to understand this potential neuronal differentiation process.

We originally examined mitosis at both P6 and P7 after Kif20a knockout, but only P7 data were included in the previous Fig. 1. As suggested, we have now examined one day earlier at P5 (Tamoxifen injection was maintained at P4, to be consistent with the scheme of tamoxifen induction used in the model of Ptc deletion-induced spontaneous medulloblastomas). The revised Fig. 1b now included data from all three stages (P5, P6 and P7), showing a loss of Ki67+ cells in the EGL of Kif20a knockouts. In addition, we also examined a later stage at P14, when the peak EGL proliferation would be mostly completed (by P15). The data from the P14 littermate brains were shown in Supplementary Fig. 1b.

2. The authors claim that by depletion of Kif20a the cells differentiate to become more mature neurons rather than going into apoptosis or cell cycle arrest. Still, there is no proof of this differentiation in Fig. 2. For example, in Fig 2b when showing NeuN staining, the authors have not quantified this or included NeuN staining of any control animals with wild-type Kif20a. Authors show NeuN staining, likely negative, in Fig 1a, which is not commented on in the results. This must be an important difference to quantify in order to understand the differentiation process. Further, Cl. Casp 3 staining of Kif20a depletion and a cell cycle analysis would be useful in order to rule out apoptosis and confirm arrest as compared to controls.

We have included comparison between control animals and Kif20a knockouts for NeuN staining. The new data were included in the revised Fig. 2b.

We have included activated caspase 3 staining comparing Kif20a knockouts and control animals at both P6 (Fig. 1c) and P7 (Supplementary Fig. 1c).

As suggested, we performed flow cytometry-based cell cycle analyses comparing Kif20a knockouts and control animals. Our data showed that knockouts and controls had comparable amount of DNA contents at 4N, suggesting that there was no obvious cytokinesis defect in the knockouts. The new data were shown in Fig. 1d.

3. The authors claim that depletion of Kif20a also in MB promotes differentiation of nascent progenitors. Still, again one would expect an increase in the number of differentiated cells in these animals, like an increased number of NeuN positive cells, or similar.

We have included comparison between single Ptc knockouts and double Ptc/Kif20a knockouts for NeuN staining. The new data were consistent with induced differentiation of cancerous GNPs and were included in the revised Fig. 4b.

4. Is increased KIF20A a marker of poor prognosis or at least elevated in SHH MB? Further is it correlating with Ki67 levels in such tumors? There is plenty of published expression data from patients available to understand and at least report this data here.

We analyzed KIF20A expression in patient transcriptome datasets with the help of Dr. Paul Northcott's group. The results showed that strong KIF20A expression is positively correlated with Ki67 level in different subgroups or subtypes of MBs. These new data were included in the revised Fig. 6a-c.

5. The mechanism of Kif20a as crucial in balancing symmetric versus asymmetric cell division in these cells is referred to and claimed in both introduction and discussion but never really examined in the

paper. Why did authors not show any proof of this using markers after division (e.g. NUMB, TRIM3) or checking correlation with the previously studied SEPT7 protein or similar.

In our previous studies of radial glial cells (RGCs) in the developing mouse cortex, we have experienced that expression of many markers, e.g. Numb or Par proteins, were not good predictor of the proliferative vs. differentiative fate of daughter cells (or symmetric vs. asymmetric divisions) of RGCs. In the case of KIF20A or SEPT7, since their presence in the intercellular bridge was transient during cytokinesis, whether they can be used as a cell division marker is not clear. Thus in all our current studies, we have focused to assess symmetric versus asymmetric divisions by characterizing phenotypic changes of NPCs on proliferation vs. differentiation, because in essence, symmetric divisions underlie progenitor cell proliferation and asymmetric divisions result in differentiation. As we discussed below, we found that the phenotypic characterization approach correlated well with monitoring cell division modes directly by live cell imaging.

In our published study of KIF20A (Geng et al., Nature Comms. 2018), we used two assay systems to examine the role of KIF20A in cell division mode control in cortical RGCs:

- (1) We first looked at functional/phenotypic changes in the knockouts. This was done either by in utero electroporation-based Cre expression in the *Kif20a^{fl/fl}* brains or by inducible knockout in the *Nestin-CreER, Kif20a^{fl/fl}* mice. Under both inducible knockout schemes, mutant cortical RGCs left the cell cycle and differentiated into neurons, which was evident by waves of new neurons migrating out of the ventricular zone into the intermediate zone and cortical plate. These data supported the in vivo function of KIF20A in balancing proliferation (symmetric divisions) and differentiation (asymmetric divisions).**
- (2) Next, to directly probe the function of KIF20A in cell divisions, we performed live cell imaging studies on cortical RGCs derived from the *Dcx-mRFP* transgenic reporter mice. Our data showed that knockdown of *Kif20a* in these RGCs led to more differentiative mode of divisions (generating neuronal progeny that are marked by expression of endogenous mRFP), thus providing direct evidence of KIF20A function in cell division mode regulation.**

In addition, the same two assay systems (genetic phenotypic analysis and live cell imaging analysis) were similarly used in our study of SEPT7 and data from both types of analyses agreed well with each other in the case of SEPT7.

Therefore we feel that, until reliable mammalian markers are available for labeling symmetric or asymmetric cell fate between the two daughter cells during an NPC division, examination of phenotypic change of NPCs in the balance between proliferation and differentiation, as we also did in this study, serves as the reliable functional criterion for determination of daughter cells' fate change or cell division mode change.

Minor

1. Tumors escape *Kif20a* depletion as these mice still die from large tumors. Is there any difference in cytokinesis or *Kif20a*-dependent intracellular transportation in these tumors as compared to *Kif20a* wild type tumors? Would the cytokinesis function of *Kif20a* in these tumor cells be compensated to a large extent by the homologous protein *Kif20b* or by other mechanisms?

We performed FACS-based cell cycle analysis for the tumor cells. Our data (Supplementary Fig. 3) showed that the two types of tumor cells had comparable 4N DNA contents, suggesting there was little difference in their cytokinesis. It is very likely that

Kif20a's cytokinesis function was compensated by Kif20b in the knockout GNPs or the tumor cells.

2. The authors suggest that SHH MB could be potentially targeted by a drug inhibiting Kif20a. Still it is not sure if any pathway in or downstream of the SHH pathway is affected by this depletion? This could then perhaps be coupled to early differentiation of GNPs. Even if such a SHH marker might be difficult to test in DAOY cells that are not considered to faithfully mimic SHH MB cells, I guess it would be easier and more relevant to find this in the presented mouse model where at least downstream *Mycn* levels would likely decrease.

As suggested, we examined the *Mycn* level in the mouse model. We performed qPCR of *Mycn* using RNAs isolated from the *Ptc* single and *Ptc/Kif20a* double knockout cerebellums at P6 and P7. The results (attached below) showed that there was a subtle increase of *Mycn* level after Kif20a knockout at both stages. However, this change might not reflect the *Mycn* level in the GNPs due to the bulk RNAs comprising both progenitor cells and progeny neurons or glia isolated from the cerebellums. Given that KIF20A and its interacting proteins (RGS3 and SEPT7) appear to function in cell division control during late cytokinesis, it is likely that KIF20A may not directly regulate the pathway induced by SHH, but perhaps have an impact on the cell division machinery activated by SHH pathway.

REVIEWERS' COMMENTS:

Reviewer #1 (Remarks to the Author):

In the revised manuscript, Qiu et al. has addressed my concerns from the previous round of review, with the minor exception of the bar graphs in Fig. 5, which still lack individual data points. These should be added prior to publication.

Reviewer #2 (Remarks to the Author):

I am happy to see that most of my concerns were carefully addressed or clarified by the authors. The inclusion of additional time points for tamoxifen treatment looks more convincing. Great to see more clearly that differentiation rather than apoptosis/cytokinesis defects played a role in the Kif20a depletion. It is also evident that KIF20A has a role in promoting SHH tumor proliferation not only in mouse models but also in patients. Although significant as it is, it seems like KIF20A in Figure 6c shows even better correlation with KI67 in SHH tumors as compared to tumors of other subgroups. Overall the paper is much improved and I have no further comments.

Responses to Reviewers' Comments

Reviewer #1 (Remarks to the Author):

In the revised manuscript, Qiu et al. has addressed my concerns from the previous round of review, with the minor exception of the bar graphs in Fig. 5, which still lack individual data points. These should be added prior to publication.

We apologize for having missed Figure 5. We have revised it with integrated data points in all graphs.

Reviewer #2 (Remarks to the Author):

I am happy to see that most of my concerns were carefully addressed or clarified by the authors. The inclusion of additional time points for tamoxifen treatment looks more convincing. Great to see more clearly that differentiation rather than apoptosis/cytokinesis defects played a role in the Kif20a depletion. It is also evident that KIF20A has a role in promoting SHH tumor proliferation not only in mouse models but also in patients. Although significant as it is, it seems like KIF20A in Figure 6c shows even better correlation with KI67 in SHH tumors as compared to tumors of other subgroups. Overall the paper is much improved and I have no further comments.

Thank you for the positive comments.

It is very possible that KIF20A may be more important for certain types or subtypes of tumors. KIF20A may be one of the essential kinesin motors (including perhaps KIF20B and other mitotic kinesins) in a general mechanism of cell fate determination and each kinesin component may have a different weight in different tumor (stem/progenitor cell) systems.